# Polyamine Metabolism in *Leishmania* Parasites: A Promising Therapeutic Target

**DOI:** 10.3390/medsci10020024

**Published:** 2022-04-22

**Authors:** Nicola S. Carter, Yumena Kawasaki, Surbhi S. Nahata, Samira Elikaee, Sara Rajab, Leena Salam, Mohammed Y. Alabdulal, Kelli K. Broessel, Forogh Foroghi, Alyaa Abbas, Reyhaneh Poormohamadian, Sigrid C. Roberts

**Affiliations:** School of Pharmacy, Pacific University Oregon, Hillsboro, OR 97123, USA; cartern@pacificu.edu (N.S.C.); kawa4755@pacificu.edu (Y.K.); nahatas@pacificu.edu (S.S.N.); elikaees@pacificu.edu (S.E.); raja2119@pacificu.edu (S.R.); sala8601@pacificu.edu (L.S.); alab5553@pacificu.edu (M.Y.A.); broe6652@pacificu.edu (K.K.B.); foro9462@pacificu.edu (F.F.); abba9206@pacificu.edu (A.A.); poor1405@pacificu.edu (R.P.)

**Keywords:** *Leishmania*, polyamines, putrescine, spermidine, ornithine decarboxylase, *S*-adenosylmethionine decarboxylase, spermidine synthase, drug resistance, transport

## Abstract

Parasites of the genus *Leishmania* cause a variety of devastating and often fatal diseases in humans and domestic animals worldwide. The need for new therapeutic strategies is urgent because no vaccine is available, and treatment options are limited due to a lack of specificity and the emergence of drug resistance. Polyamines are metabolites that play a central role in rapidly proliferating cells, and recent studies have highlighted their critical nature in *Leishmania*. Numerous studies using a variety of inhibitors as well as gene deletion mutants have elucidated the pathway and routes of transport, revealing unique aspects of polyamine metabolism in *Leishmania* parasites. These studies have also shed light on the significance of polyamines for parasite proliferation, infectivity, and host–parasite interactions. This comprehensive review article focuses on the main polyamine biosynthetic enzymes: ornithine decarboxylase, *S*-adenosylmethionine decarboxylase, and spermidine synthase, and it emphasizes recent discoveries that advance these enzymes as potential therapeutic targets against *Leishmania* parasites.

## 1. Introduction to *Leishmania*

Leishmaniasis is a neglected tropical disease that is endemic in nearly 90 countries across Africa, Asia, the Middle East, Europe, and Central and South America. More than 1 billion people are impacted by leishmaniasis worldwide with approximately 1 million new cases and 70,000 deaths occurring each year [1,2,3]. Poverty is a major risk factor underlying this disease, with increased human migration, civil war, and unrest contributing to recent outbreaks [2,3,4,5]. Furthermore, environmental damage such as deforestation, urbanization, and climate change have also contributed to a worldwide increases in leishmaniasis [2,5].

*Leishmania* parasites undergo two major morphological changes as they shuttle between the sandfly and mammalian host. The flagellated promastigote form resides in the gut of the sand fly, and the non-motile amastigote form inhabits the acidic phagolysosome within macrophages in the mammalian host. Transmission to the mammalian host occurs when an infected sand fly takes a bloodmeal. Parasites are phagocytosed by neutrophils, macrophages, or other antigen-presenting cells, although macrophages are the ultimate host cell [6,7,8]. Phagosomes containing internalized parasites merge with lysosomes to form phagolysosomes. Here, changes in temperature between the macrophage and sandfly, as well as the acidic pH of the phagolysosome, trigger the conversion of promastigotes to amastigotes and adaptation to the hostile host environment [9,10,11].

There are more than 20 species of *Leishmania* that infect humans and cause a spectrum of clinical symptoms. Visceral leishmaniasis (VL) is caused by *L.*
*donovani* and *L.*
*infantum* (*L.*
*chagasi*) and is nearly always fatal if left untreated. It affects internal organs such as the spleen, liver, and bone marrow, with symptoms that include hepatosplenomegaly, fever, and weight loss. *L. major*, *L. tropica*, and *L. mexicana* are the main *Leishmania* species that cause cutaneous leishmaniasis (CL), which forms ulcerative skin lesions. A less common manifestation is mucocutaneous leishmaniasis (MCL), which may be caused by *L.*
*amazonenis* and *L.*
*braziliensis* and results in both sores and the destruction of mucous membranes in the nose, mouth, or throat [3,12].

Vaccines for preventing leishmaniasis in humans are not available, although recent clinical trials have shown some promise [13]. While three canine vaccines have been approved, these only provide partial protection [13,14]. Thus, drug therapy is crucial, but the currently available drugs for treating leishmaniasis are limited in number, have severe adverse effects, and are becoming sidelined by drug resistance [13]. Pentavalent antimonials were introduced over six decades ago, and despite significant and severe side effects associated with their administration, and widespread resistance to their antileishmanial effects, they are still used as a first-line treatment in many countries. Their mechanism of action is complex, although the inhibition of trypanothione reductase and disruption of trypanothione metabolism appears to contribute significantly to their mode of action [13,15]. Amphotericin B, an antifungal medication, binds to ergosterol that is present in the membranes of *Leishmania* and related parasites, as well as susceptible fungi, but it is absent in the human host. Amphotericin B is available as a deoxycholate salt, which causes nephrotoxicity and other significant side effects, or as liposomal formulation that has fewer associated adverse effects and improved pharmacokinetic properties, but it is expensive unless an access price has been negotiated [16]. Miltefosine is the only orally administered treatment approved for leishmaniasis. It appears to have multiple antileishmanial effects, including interference with phospholipid metabolism and the disruption of mitochondrial and acidocalcisome function [13,15,17]. Widespread use of miltefosine for treating leishmaniasis is limited by its teratogenic properties (the drug is contraindicated for use in pregnant women or in those that may become pregnant) and due to the emergence of drug-resistant strains [13,15,18]. Paromomycin, an aminoglycoside antibiotic, has been utilized with some success against VL and CL, but again, side effects such as hepatotoxicity occur and drug resistance have been demonstrated in the laboratory [13,15]. Pentamidine, an aromatic diamidine, is a second-line treatment that appears to act by accumulating in the mitochondria and binding to kinetoplastid DNA and topoisomerase II [13]. Combination therapy to reduce drug resistance and treatment failure have recently been attempted and met with mixed success. Clinical studies combining miltefosine with paromomycin or amphotericin B or combinations of antimonials with allopurinol were promising, while others found little advantage compared to monotherapy [13,19,20]. Further complicating therapeutic management is that *Leishmania* parasites may never disappear after clinical treatment but remain dormant at low numbers and can be reactivated [21,22,23,24]. This observation has stimulated recent interest in understanding the mechanisms that allow parasites to persist [21,25,26,27].

Thus, there is an urgent need for the identification and validation of new therapeutic targets. The polyamine biosynthetic pathway may be such a target. Polyamines are essential for *Leishmania*, and the parasite metabolic pathway shows significant differences to that of the mammalian host [28,29,30,31,32,33]. Furthermore, the polyamine pathway has been validated as a clinical target in the related parasite, *Trypanosoma brucei* [32,34,35,36,37].

## 2. Significance of Polyamines

The polyamines putrescine, spermidine, and spermine are small organic cations containing several amine groups that are positively charged under physiological conditions (Figure 1). Polyamines are ubiquitous and play critical roles in a variety of key processes, including growth, differentiation and macromolecular synthesis. Because of their relevance in rapidly proliferating cells, they have long been of interest in cancer and parasite research [32,33,38,39,40,41,42]. Furthermore, polyamines have shown protective properties, including anti-oxidant, anti-aging, and cardio- and neuroprotective functions [42,43,44,45,46]. In contrast, their catabolic by-products have been associated with tissue damage and diseases such as cancer, heart disease, and kidney failure [38,39,42,47,48,49]. Polyamine homeostasis is intricately regulated in mammalian cells and imbalances, such as excess polyamine production, have been associated with cancer and inflammatory disorders [38,42,45,46]. While the function and metabolism of polyamines have been extensively studied, particularly in the mammalian system, questions still remain about the spectrum of their effects in cells [42,45,46,50]. Despite the simple chemical structure of polyamines, they engage in complex interactions with cellular components and processes, which can be both specific and non-specific in nature [51]. Even less is known about the function or regulation of polyamines in parasites, despite their significance in parasite biology. This is underscored by recent studies highlighting their importance as potential therapeutic targets in trypanosomatids, which include *Leishmania* spp., *T. brucei*, and *Trypanosoma cruzi* [30,32,33,52,53,54]. The polyamine pathway in *T. brucei*, the causative agent of African sleeping sickness, is the target of the drug D, L-α-difluoromethylornithine (DFMO, eflornithine, ornidyl), an inhibitor of ornithine decarboxylase (ODC), that has shown remarkable therapeutic efficacy in the treatment of African trypanosomiasis [32,34,35,36,37].

The polyamine pathway of mammalian cells and *Leishmania* is depicted in Figure 2. The pathway can be primed by the enzyme arginase (ARG), which converts arginine to ornithine. In mammalian cells, ODC is considered to be the first and a rate-limiting enzyme, and it decarboxylates ornithine to form putrescine. Spermidine synthase (SPDSYN) and spermine synthase (SPMSYN) sequentially form spermidine and spermine by adding an aminopropyl group. The aminopropyl group is donated by decarboxylated *S*-adenosylmethionine, which is synthesized by *S*-adenosylmethionine decarboxylase (ADOMETDC). Spermine can be back-converted to spermidine and putrescine by the concerted action of spermidine/spermine N1-acetyltransferase and N1-acetylpolyamine oxidase. Spermine can also be converted to spermidine by spermine oxidase. In both scenarios, toxic side products such as hydrogen peroxide and reactive aldehydes are formed. A crucial downstream reaction of the polyamine pathway is the hypusination of eukaryotic translation initiation factor eIF5A, which requires spermidine for its activation.

The polyamine biosynthetic pathways of trypanosomatids are significantly different from that of the mammalian host [32,33,55] (Figure 2). For example, spermidine and spermine predominate in mammalian cells, while putrescine and spermidine are more abundant in trypanosomatids and other single cell organisms [56]. The polyamine spermine is neither produced nor utilized in trypanosomatids and the back-conversion of spermidine to putrescine does not exist [57,58,59]. Furthermore, while ODC and ADOMETDC are rapidly turned over in mammalian cells, these enzymes have a much longer half-life in trypanosomatid parasites [32,33,60,61,62]. Indeed, the selectivity of DFMO toward *T. brucei* is not due to disparate binding affinities to the parasite versus mammalian ODC enzymes, but rather that DFMO, as a covalently bound inhibitor, incapacitates the long-lived parasite ODC, while the human enzyme is rapidly replenished by resynthesis [61,63]. The trypanosomatid ADOMETDC is activated by the formation of a heterodimer with prozyme, which is a catalytically dead paralog of the parasite ADOMETDC. In a reaction unique to trypanosomatids, spermidine is conjugated to two glutathione molecules to produce trypanothione, which is the major intracellular thiol in *Leishmania* and other trypanosomatids and is essential for maintaining the intracellular redox balance and in oxidant defense [31,64,65]. The hypusination and activation of eIF5A occurs in both trypanosomatids and mammalian cells, although the enzyme deoxyhypusine synthase shows unique structural features in trypanosomatids [32,66,67,68].

Differences in the polyamine biosynthetic pathways also exist between the trypanosomatid parasites (Figure 3). In the mammalian host, *Leishmania* parasites live in phagolysosomes of macrophages, *T. cruzi* reside in the cytosol of host cells, and *T. brucei* are present in the bloodstream. In *Leishmania* parasites, the enzyme arginase (ARG) is considered to be the first enzyme of the polyamine biosynthetic pathway because the only essential role of ornithine is as precursor for polyamine biosynthesis [69,70,71,72]. Arginine is an essential amino acid and needs to be scavenged from the host for polyamine synthesis. *Leishmania* parasites contain ODC, SPDSYN, and ADOMETDC enzymes, while SPMSYN and back-conversion enzymes are absent. *T. brucei*, an extracellular parasite, contains an inactive ARG analog, and although the conversion of arginine to ornithine appears to occur, the majority of ornithine is salvaged from the host [32,73,74]. Because the levels of salvageable putrescine and spermidine in the bloodstream are too low (<1 µM) to meet the growth needs of *T. brucei*, the inhibition of ODC by DFMO is an effective therapeutic strategy for depleting putrescine in these parasites [32,75]. The polyamine pathway in *T. cruzi* parasites is substantially abbreviated. These parasites lack ARG and ODC [32,33,76], and they are dependent on putrescine salvage for growth and viability. They do, however, contain functional SPDSYN and ADOMETDC enzymes [77,78]. 

Because polyamines are essential for *Leishmania* parasites and their metabolism shows significant differences to that of the mammalian host, the polyamine pathway offers several possibilities for therapeutic design. While several recent review articles have focused on the therapeutic potential of *Leishmania* arginase [52,54,79,80] and trypanothione metabolism [31,64,81,82,83], the intent of this review is to focus on ODC, ADOMETDC, and SPDSYN in *Leishmania* and discuss their promise as therapeutic targets.

## 3. Relevance of Polyamines for Host Parasite Interactions

*Leishmania* amastigotes reside within phagolysosomes in host macrophages. Studies with intracellular amastigotes, especially in intralesional macrophages, are difficult to perform, and thus, many questions remain about the host nutrient environment, as well as parasite nutrient requirements. It is well established that *Leishmania* are auxotrophic for purines, heme, certain vitamins, and several amino acids, and the supply of these essential nutrients is important for supporting parasite growth. Amino acids, sugars, and lipids are believed to be available in phagolysosomes, and it is probable that an array of host nutrients is delivered to the phagolysosome via fusion with phagocytic and endocytic vesicles [84,85,86,87,88]. Overall, a complex picture is emerging where the phagolysosome is both rich in some nutrients and limited in other metabolites [86].

*Leishmania* parasites modulate the host immune response and metabolism to favor their own survival [7,8,88,89,90]. Arginine, an essential amino acid, is required for polyamine biosynthesis, and *Leishmania* parasites compete with host macrophages for this metabolite [91,92,93]. In macrophages, arginine is a key substrate for two competing pathways: it can be converted to ornithine by ARG or alternatively into the potent antileishmanial agent nitric oxide by inducible nitric oxide synthase (iNOS) [80,94,95]. Two types of ARG are present in mammalian cells: type I, which is located in the cytosol, and type II, which is present in mitochondria. Numerous experiments and clinical observations have correlated an increased host ARG I expression and activity with augmented parasite loads and thus firmly established the mammalian ARG I as a key factor for *Leishmania* infections [80,96,97,98,99,100,101,102]. While it is not completely understood how increased levels of host ARG I contribute to disease exacerbation, one effect of higher ARG I activity may be the depletion of arginine levels, which would reduce the production of nitric oxide by iNOS. A local reduction in arginine has also been shown to impair the development of T cells, leading to suppression of the immune response and increased parasitemia [103,104,105]. It has furthermore been speculated that increased host ARG I activity leads to higher levels of host polyamines, which could be subsequently scavenged by intracellular parasites [87,94,95,99,106]. However, there is no direct evidence that *Leishmania* parasites salvage polyamines from the host, and murine infectivity studies with *L. donovani* polyamine pathway gene deletion mutants suggest otherwise (described below) [107,108].

There is only one ARG in *Leishmania*, and *ARG* gene deletion mutants have been generated in *L. mexicana*, *L. major*, *L. amazonensis*, and *L. donovani* [69,70,71,72]. *ODC* and *SPDSYN* gene deletion mutants have been created in *L. donovani* [107,108]. These mutants are valuable tools for assessing whether ornithine or polyamines are scavenged by intracellular parasites. Infectivity studies in mice demonstrated that the deletion of *ARG* caused reductions in infectivity compared to wild-type parasites in all four species investigated; however, infections were still established by these mutants [69,70,71]. In contrast, the *L. donovani ODC* and *SPDSYN* knockout parasites revealed profound reductions in infectivity [107,108]. The most dramatic drop of infectivity was observed with the *L. donovani ODC* gene deletion mutants, which showed an infectivity six orders of magnitude lower than wild-type parasites [107]. Similarly, the *SPDSYN* knockout strains exhibited a profound deficit in infectivity [108], and murine infectivity studies have also shown that *L. donovani ADOMETDC* gene deletion mutants exhibited a reduced infectivity phenotype (Buddy Ullman, personal communication).

The intriguing variations in intracellular survival across the various *L. donovani* mutants may be due to differences in the levels of salvageable ornithine versus polyamines in the phagolysosome. Accordingly, a model was proposed in which spermidine and especially putrescine availability is severely limited in macrophages under physiological conditions [69] (Figure 4). In support of this model, labeled arginine was rapidly converted to spermine (which parasites cannot utilize), and the levels of ornithine and spermine were increased in bone marrow-derived macrophages infected with *L. major* [99]. In contrast, only negligible amounts of putrescine and spermidine could be measured [99]. These results are consistent with other studies on differentiated mammalian cells (such as macrophages), which typically contain low levels of putrescine and spermidine [109,110].

Several studies indicate that when putrescine is supplied exogenously, it can be accessed by intracellular amastigotes within the phagolysosome. The addition of putrescine to the drinking water of mice infected with *L. donovani ODC* gene deletion mutants partially reversed the avirulent phenotype of the mutants [111]. Similarly, the addition of ornithine or putrescine to the media of infected macrophages increased the number of intracellular *L. mexicana* wild-type parasites and *ARG* gene deletion mutants [112]. These data suggest that it is possible that polyamines are normally limited under physiological conditions but that diets rich in polyamines may transiently increase their levels within infected cells. Additional research into determining polyamine levels in regular and stimulated macrophages, and particularly within the phagolysosome of infected cells, will shed light on the accessibility of host polyamines for *Leishmania.* Overall, however, the dramatically reduced infectivity phenotypes of *L. donovani ODC*, *SPDSYN,* and *ADOMETDC* gene deletion mutants bolsters these enzymes as promising therapeutic targets. In the following sections, we will review these enzymes with an emphasis on their structure, function, and inhibition profiles.

## 4. Ornithine Decarboxylase (ODC)

### 4.1. Enzyme Structure and Function

In 1992, Hanson et al. identified the *leishmanial ODC* gene in *L. donovani*, encoding a protein of 707 amino acids [113]. Since then, *ODC* sequences from several *Leishmania* species have been sequenced [114,115]. The ODC amino acid sequences of *L. donovani* (LdBPK_120105.1), *L. major* (LmjF.12.0280), and *L. braziliensis* (LbrM.12.0300) contain 707, 707, and 636 amino acids, respectively, which is significantly more than the *T. brucei* (Tb927.11.13730), mouse (NP_038642.2), or human (AAA59967.1) ODC with 423, 461, and 461 amino acids, respectively. This discrepancy is primarily due to a unique N-terminal extension in the *Leishmania* ODC sequence consisting of an extra ≈250 amino acids [113,116]. The removal of this extension resulted in increased stability of the enzyme but a loss of enzymatic activity due to improper folding [116]. The unique N-terminal extension and its importance for enzyme activity presents an opportunity for the development of inhibitors specific to the *leishmanial* ODC. Alignments with Needle (EMBOSS) or Stretcher (EMBOSS) that provide global alignments of the entire sequence, including the N-terminal extension, show identities of only ≈24% between leishmanial and mammalian ODCs (Table 1), while programs that optimize for partial sequence alignments find ≈40% identity (for example, Clustal Omega or NBCI blast). A three-dimensional amino acid structure model of an *L.*
*donovani* ODC alignment of 368 residues showed 45% identity with the human ODC [117]. Interestingly, the *T. brucei* ODC lacks the N-terminal extension found in *Leishmania* and is more similar to the human and mouse ODC with ≈60% identity (Table 1).

The kinetic properties of the *Leishmania*, *T. brucei*, mouse, and human ODC have been characterized and established. Both the binding affinity to ornithine and the catalytic efficiency is somewhat higher in the human ODC compared to the leishmanial enzyme [118,119,120]. The K_m_ for ornithine is 0.08 and 0.39 for the human and *L. donovani* ODC, respectively, and the catalytic efficiency (K_cat_/K_m_) is 41.3 and 20.8 for the human and *L. donovani* ODC, respectively [118]. The crystal structures for the mouse, human, and *T. brucei* ODC proteins in the presence and absence of inhibitors have been solved [32,118,121,122,123,124], and the *leishmanial* ODC (which lacks a crystal structure) has been modeled against these structures.

One significant difference between the trypanosomatid and mammalian ODCs is in their stability. Mammalian ODCs have a high turnover rate with half-lives between 5 and 30 min [125], while the leishmanial ODC is stable for over 20 h [113] and the *T. brucei* ODC is stable for over 6 h [63]. Similar to the *T. brucei* ODC, the leishmanial enzyme lacks the COOH terminus that is responsible for the rapid degradation of the mammalian enzyme [113,126]. The slow turnover rate of *Leishmania* ODC in addition to the unique structural feature of the N-terminal extension makes this enzyme a prime candidate for drug targeting.

### 4.2. ODC Gene Deletion Studies

Deletion of the *ODC* gene in *L. donovani* led to growth arrest in promastigotes, axenic, and intracellular amastigotes, and it profoundly reduced infectivity in mice [57,107,111,127]. When *ODC* gene deletion mutants were incubated in polyamine-free media, intracellular putrescine pools were rapidly depleted while levels of spermidine, though initially decreased, were sustained at stable levels for over 12 days [57,127]. Although it remains unclear how spermidine levels were maintained in these mutants, it is possible that the back-conversion of trypanothione to spermidine via the bifunctional trypanothione synthetase-amidase helps to maintain intracellular spermidine levels. Alternatively, spermidine pools may be conserved by producing less hypusinated eIF5A and trypanothione. Indeed, an increase in glutathione and decline in trypanothione levels is observed in cells lacking the *ODC* gene [57]. The depletion of putrescine, spermidine, and trypanothione likely all contribute to the cell death phenotype observed in the *ODC* gene deletion mutants. The supplementation of *L. donovani ODC* or *ARG* gene deletion mutants with spermidine or spermine did not restore growth, demonstrating that putrescine is not merely a precursor for the production of spermidine as previously postulated [29,57,71] but has additional essential roles [69,127].

This observation is bolstered by recent observation on polyamine-starved *ODC* and *SPDSYN* gene deletion mutants. These studies revealed that the starved *ODC* gene deletion cells, which lack putrescine, rapidly ceased proliferation after the removal of polyamine from the culture media, and they died over a course of two weeks [127]. In contrast, the starved *SPDSYN* gene deletion mutants, which accumulate putrescine, continued to proliferate for several days before entering into a quiescent-like state, and these cells could be maintained without polyamine supplementation for approximately six weeks [127]. These observations are in line with in vivo infectivity studies which indicate that the *ODC* gene deletion mutants show a more profoundly reduced infectivity in comparison to the *SPDSYN* gene mutants [107,108], implying that maintaining intracellular putrescine levels is most important for parasite proliferation and viability.

### 4.3. Efficacy of the Ornithine Analog DFMO as ODC Inhibitor

The fluorinated ornithine analog and ODC inhibitor DFMO (Figure 5) is an approved drug to treat hirsutism and trypanosomiasis and is being explored for the prevention and treatment of cancer in clinical trials [1,34,35,36,37,38,39,40,128,129,130,131,132].

Because of the success of DFMO against African trypanosomes, it is not surprising that many studies have investigated the efficacy of the ornithine analogue against *Leishmania* parasites. Multiple studies have demonstrated that DFMO inhibits the growth of *Leishmania* promastigotes and axenic amastigotes (Table 2). However, discrepancies of DFMO efficacy among *Leishmania* species exist, with *L. donovani* and *L. infantum* promastigotes being highly susceptible to DFMO, while *L. mexicana* and *L major* are resistant. The mechanism of action of DFMO is specific to the inhibition of ODC as the enzyme activity was inhibited and putrescine levels diminished upon treatment, while putrescine supplementation rescued growth inhibition [58,107,133,134]. In addition, the overexpression of ODC from an episome conferred resistance to DFMO [135], and laboratory-induced resistance to DFMO was accompanied by increased levels of ODC, although additional adaptations may have occurred [113,134,136,137].

Several in vivo rodent infectivity studies have been performed with DFMO administered in drinking water and show variable results (Table 3). DFMO is effective against the visceralizing species *L. donovani* and *L. infantum* in mice and Golden Hamsters, where some studies demonstrated a remarkable suppression of infectivity up to 99%, although infections were not completely cleared [111,138,139,140]. The drug is also effective against the mucocutaneous species *L. braziliensis guyanensis* [138]. In contrast, DFMO showed little efficacy against *L. mexicana* in mice [138].

Both in vivo and in vitro data show that DFMO is effective against the visceralizing species *L. donovani* and *L. infantum*, and against *L. braziliensis guyanensis* (cutaneous and mucocutaneous leishmaniasis) but not against *L. mexicana* and *L. major*, which cause cutaneous leishmaniasis [138]. Because this discrepancy in efficacy is found in both promastigotes and intracellular amastigotes in vivo, it appears to be specific to the parasite species and not the different host environments (liver/spleen versus skin). Differences in the ODC structure and active site are unlikely to cause the discrepancy in susceptibility because the primary sequences are similar (Table 1); however, data comparing the inhibition kinetics from different species are not available. Variations in polyamine transport, and thus in the capacity to salvage polyamines from the extracellular environment, between the species might contribute to the variable efficacy. Alternatively, differences in the intracellular accumulation of DFMO could also account for the disparities in susceptibility. Future studies are needed to elucidate the cause for the observed differences in efficacy.

Together, these infectivity studies show promise but also species-dependent variations, which explains conflicting statements about the potential and effectiveness of DFMO in the literature. In addition, DFMO is expensive to produce and has poor pharmacokinetic properties [37], making its clinical use as an antileishmanial drug unlikely.

### 4.4. Putrescine and Agmatine Analogs as ODC Inhibitors

The putrescine analog 3-aminooxy-1-aminopropane (APA) (Figure 5) is effective in vitro against *L. donovani* promastigotes and intracellular amastigotes with EC_50_ values of 42 µM and 5 µM, respectively (Table 4) [59]. The inhibitor is specific for ODC, as the growth inhibition of promastigotes was rescued by the addition of supplementary putrescine or spermidine or overexpression of ODC. Furthermore, APA decreased ODC activity with a concomitant reduction in putrescine, spermidine, and trypanothione levels [59]. APA is also an inhibitor of the mammalian ODC, and the crystal structure of the human ODC in complex with APA has been reported and was used for structural modeling of the leishmanial ODC–APA complex [118]. The model predicts that APA competes with ornithine for the binding site on the leishmanial ODC. APA has comparable high affinity K_i_ values for the human (1.4 nM) and leishmanial (1.0 nM) recombinant ODC. Yet, concentrations of up to 200 µM APA proved to be non-toxic to murine J774 macrophages [118].

Because APA only passively enters *Leishmania* cells, chemical modifications were carried out to improve active transport across the membrane, which resulted in the compound gamma-guanidinooxy propylamine or 1-guanidinooxy-3-aminopropane (GAPA) (Figure 5) [141,142]. While APA is a putrescine analog, GAPA is structurally closer related to agmatine, which is an aminoguanidine [141] (Figure 5). Yet, GAPA inhibited the uptake of labeled putrescine, suggesting that it is taken up via a putrescine transporter [142]. Although GAPA is less potent than APA as an inhibitor of the recombinant *L. donovani* ODC, it proved to be effective against promastigotes and intracellular amastigotes with EC_50_ values of 36 µM and 9 µM, respectively (Table 4). The improved transport of GAPA may compensate for the relatively low affinity to ODC and thus yield similar efficacy as observed with APA [141]. The observations that ODC overexpression conferred resistance to GAPA, that ODC activity and polyamine levels were markedly lower in cells treated with GAPA, and that cell proliferation in the presence of GAPA could be rescued by the addition of putrescine or spermidine, provide strong evidence that GAPA acts specifically through ODC inhibition [142]. Strikingly, although both putrescine and spermidine levels were decreased in promastigotes treated with GAPA, trypanothione levels were not impacted [142], the reasons for which remain unclear. Although APA is not toxic to J774 macrophages, such analyses have yet to be performed for GAPA. Nevertheless, GAPA remains an intriguing compound, as it inhibits both ODC activity and putrescine transport into the parasites [142], and thus should effectively prevent the replenishment of the parasite putrescine pool by salvage. Furthermore, the compound is effective against sodium stibogluconate-resistant *L. donovani* isolates [141].

A study by Vannier-Santos et al. found that the putrescine analog 1,4-diamino-2-butanone (DAB) (Figure 5) inhibited *L. amazonensis* proliferation with an EC_50_ of 144 µM and was effective against intracellular amastigotes [143]. Similar to GAPA, DAB blocks both putrescine uptake and ODC activity. However, DAB likely interacts with additional cellular targets, since only high concentrations of putrescine or spermidine (10 mM) reversed growth inhibition. DAB causes structural and functional damage to mitochondria in *Leishmania* and in *T. cruzi*, which might suggest that polyamines, and in particular putrescine, have a functional role in maintaining mitochondrial integrity [143,144]. Thus, while the complete mechanism of action of DAB has not been fully elucidated and the EC_50_ value in *L. amazonensis* promastigotes is not impressive, it may be a promising lead compound.

### 4.5. Spermine Analogs as ODC Inhibitors

The bis(benzyl)polyamine or spermine analog MDL 27695 (N,N′-bis(3-((phenylmethyl)amino)propyl)-1,7-diaminoheptane) proved to be highly effective in both mice and hamster infectivity studies with *L. donovani* [140,145,146]. Depending on the amount and type of administration of the drug, overall parasite burdens were reduced between 92 and 99.9% in livers of BALB/c mice. Remarkably, even the oral administration of MDL 27695 was effective. The outcome in the hamster model was slightly less impressive with reduced parasite burdens in the liver of around 50–77%. MDL 27695 is an ODC inhibitor and reduced both putrescine and spermidine levels in the liver and spleens of infected hamsters [140]. One study found that MDL 27695 together with CGP 40215A, an inhibitor of ADOMETDC, was more effective than either compound alone (although not additive) in inhibiting *L. donovani* promastigote proliferation [147]. MDL 27695 and several other bis(benzyl)polyamine analogs (MDL 27693, MDL 27700, and MDL 27994) were subsequently evaluated in vitro [148]. Their EC_50_ values against *L. donovani* promastigotes ranged from 4 to 25 µM [148]. The analogs effectively inhibited protein synthesis, while the inhibition of replication and transcription was less pronounced and varied among compounds. All of the analogs demonstrated significant inhibition of ODC as well as ADOMETDC activity in vitro, and putrescine and spermidine levels were depleted or at least reduced depending on the type of compound. Intriguingly, sub-inhibitory concentrations of the analogs were able to rescue *L. donovani* promastigotes from DFMO inhibition, suggesting that the compounds were able to substitute for the physiological function of putrescine and/or spermidine at least to some extent [148]. It should be noted that these studies were performed in the early 1990s, and despite the promising early results, no follow up has been reported in the literature.

### 4.6. Other Inhibitors of ODC

Structures for the human and the *T. brucei* ODC have been used for computer modeling of the *Leishmania* ODC, in silico screening, and analysis of inhibitors [117,118,149,150,151]. A di-epoxide derivative of the natural product diospyrin demonstrated non-competitive inhibition of ODC in computational docking studies and exhibited potent EC_50_ values of 2.7 µM and 0.18 µM against *L. donovani* promastigotes and intracellular amastigotes, respectively [152]. Treatment of *L. donovani* infected BALB/c mice with 2 mg/kg/day of the di-epoxide derivative of diospyrin; however, it showed only a modest reduction of 38% of the hepatic parasite load [152]. Interestingly, studies demonstrated that diospyrin is a specific inhibitor of DNA topoisomerase I [153], while computational analysis of the di-epoxide derivative suggests that it binds to ODC [152]. Diospyrin and its derivatives were shown to induce mitochondrial membrane depolarization and induce apoptosis [154].

A computational approach was used to screen over 160,000 natural compounds, which were obtained through the free ZINC database (https://zinc.docking.org/ accessed on 1 April 2022), against the predicted structures of ODC and SPDSYN [150]. This yielded two natural compounds (N-[1] benzofuro [3, 2-d] pyrimidin-4-yl-d-tryptophane (BFPT) and dihydrocitrinone (DHC)) that were predicted to inhibit both enzymes [150]. Similarly, one study reported the virtual screening of over 130,000 compounds of the ZINC database, which produced 12 compounds with good ODC inhibition profiles, with two of the chemicals showing promise after virtual ADME and toxicity evaluations [117]. Yet another in silico screen of the database identified three inhibitors of ODC, N-[4-(2-oxo-2Hchromen-3-yl)phenyl]-1H-1,2,4-triazole-3-carboxamide (M-2), 8-[3-(2,5-dimethylpyrrol-1-yl)benzoyl]-3-(4-methoxyphenyl)-1-oxa-8-azaspiro [4.5]dec-2-ene (M-5), and 1,3,6,7-tetrahydroxyxanthone C2-β-D-glucoside (mangiferin) that were further characterized [116]. The compounds are uncompetitive (M-2) or non-competitive inhibitors (M-5, mangiferin) with K*i* values between 78 and 371 µM against recombinant *L. donovani* ODC. Efficacy against *L. donovani* promastigotes was moderate with EC_50_ values of 350 µM, 125 µM, and 950 µM for M-2, M-5, and mangiferin, respectively. Putrescine and spermidine levels were reduced by ≈40% and ≈20%, respectively. Treatment of parasites with the novel ODC inhibitors caused an increase in SPDSYN mRNA levels in parasites, which may explain the only modest reduction in spermidine pools. Consistent with this observation, the inhibition of ODC by APA or the deletion of the ODC gene also only modestly reduced spermidine levels [57,59,127]. This might suggest that either ODC activity or intracellular putrescine levels may help regulate SPDSYN expression.

## 5. *S*-adenosylmethionine Decarboxylase (ADOMETDC)

### 5.1. Enzyme Structure and Function

The first ADOMETDC genes were identified in *L. donovani* and *L*. *infantum* in 2002 [155,156], and sequences from several *Leishmania* species are now available [114,115]. The amino acid sequences of ADOMETDC from *L. donovani*, *L. infantum*, *T. brucei*, and *T. cruzi* are similar with identities between 61.8% and 97.6% (Table 5). The *Leishmania* ADOMETDC sequences show only ≈26% identity to the human ADOMETDC sequence (NP_001625.2) when analyzed with a global alignment tool (Table 5). The mammalian ADOMETDC is also slightly smaller with 38 kDa compared to the *Leishmania* ADOMETDC proteins of 42–44 kDa.

In addition to the ADOMETDC enzyme, trypanosomatids express an ADOMETDC paralog or prozyme that is important for the enzymatic activity of ADOMETDC [32,55,157,158,159,160]. Prozyme lacks key catalytic residues, does not undergo processing, and is enzymatically inactive. ADOMETDC that is not in complex with prozyme has little activity, but the ADOMETDC/prozyme heterodimer has full enzymatic activity [68,157,160]. Indeed, ADOMETDC enzymatic activity increased by 1200-fold in *T. brucei* upon heterodimer formation [160]. Both ADOMETDC and prozyme are essential for survival in *T. brucei*, highlighting the metabolic significance of prozyme [157]. Characterization of prozyme and the ADOMETDC/prozyme heterodimer has been focused on *T. cruzi* and *T. brucei*, although *Leishmania* spp. also express an ADOMETDC paralog [159], and *L. donovani* ADOMETDC, when not in complex with prozyme, has very little enzymatic activity (Buddy Ullman, personal communication). However, more studies are needed to elucidate the function and importance of prozyme for *Leishmania*. Because the ADOMETDC prozyme is present in trypanosomatids but not in mammalian cells, prozyme and the heterodimer are promising drug targets. The X-ray structures of the *T. brucei* ADOMETDC as homodimer and as ADOMETDC/prozyme heterodimer have been solved [32], which can guide drug screening efforts. Amino acid identities between the trypanosomatid prozyme sequences are around 40% (Table 6). In contrast, the percent identity between ADOMETDC and prozyme within each species is relatively low, around 25% (Table 7).

The enzyme ADOMETDC is produced as a proenzyme that undergoes auto-processing to be enzymatically active. The cleavage produces pyruvate, which remains covalently bound as an essential cofactor for the decarboxylation reaction [161,162,163]. ADOMETDC expression in mammals is highly regulated by polyamine levels in the cell and putrescine stimulates both the cleavage and the enzymatic activity of the enzyme [60,163,164,165]. The ADOMETDC cleavage and activation site, Glu-Ser-Ser, is conserved among trypanosomatid ADOMETDC sequences [166,167]; however, unlike in mammalian cells, putrescine does not stimulate ADOMETDC processing in trypanosomatids. Putrescine stimulates the activity of the ADOMETDC/prozyme complex in *T. cruzi* but not in *T. brucei* [168]. The expression of *T. brucei* prozyme is regulated by levels of ADOMETDC protein and the ADOMETDC product decarboxylated *S*-adenosylmethionine [169,170]. These studies demonstrate that the regulation of both the expression and activity of the ADOMETDC/prozyme complex exists in trypanosomatids, although similar analyses have not yet been performed in *Leishmania* parasites. The differences in regulation of the mammalian and trypanosomatid ADOEMTDC may offer avenues for drug development.

Computational studies and biophysical analyses suggest that *L. donovani* ADOMETDC and SPDSYN form a metabolon enzyme complex [171]. These types of temporary enzyme complexes are believed to increase catalytic rates through effective substrate channeling. Although in silico modeling implies that human ADOMETDC and SPDSYN may also interact, the complex of the leishmanial enzymes is predicted to be more stable, and thus, targeting the leishmanial metabolon complex may be a promising therapeutic strategy [171]. Stability of the metabolon complexes may also be influenced by the differences in the half-life of the parasite and mammalian enzymes. The mammalian ADOMETDC has a rapid turnover rate with a half-life of less than 1 h and is degraded by the 26S proteasome [60,62]. In contrast, the *L. donovani* ADOMETDC is stable with a half-life of up to 24 h [155]. This difference in half-life may provide a therapeutic opportunity, since an irreversible inhibitor that targets ADOMETDC is likely to be substantially more effective against the long-lived leishmanial enzyme, as illustrated by the differential effect of DFMO inhibition on the mammalian and *T. brucei* ODCs.

### 5.2. ADOMETDC Gene Deletion Studies

Gene deletion studies in *L. donovani* established that ADOMETDC is an essential component of the polyamine pathway [155]. The ADOMETDC gene deletion mutants were unable to survive unless spermidine was added to the media or promastigotes were transfected with an episomal ADOMETDC construct. Putrescine and spermine were not able to restore growth. As expected, the mutants accumulated putrescine and glutathione with a concomitant decrease in spermidine and trypanothione levels [155]. Murine infectivity studies have been performed with ADOMETDC gene deletion mutants and showed reduced infectivity phenotypes compared to wild-type parasites (Buddy Ullman, personal communication).

### 5.3. Inhibitors of the Leishmanial ADOMETDC

When ADOMETDC was overexpressed in wild-type *L. donovani* promastigotes, cells were able to evade death from 5-(((Z)-4-amino-2-butenyl)methylamino)-5-deoxyadenosine (MDL 73811), an inhibitor of ADOMETDC, demonstrating that this compound is toxic and its mechanism of action is specific to the inhibition of ADOMETDC [135,147]. Methylglyoxal bis(guanylhydrazone (MGBG), CGP 40215A (a diamidine and bicyclic analog of MGBG), and diminazene aceturate (berenil) reduced ADOMETDC activity, as well as intracellular spermidine levels, and parasite growth inhibition by these compounds could be partially restored by spermidine supplementation [172]. However, ADOMETDC overproducer strains were still susceptible to berenil and MGBG [135], suggesting that the toxicity of these compounds is not solely due to ADOMETDC inhibition. The co-administration of CGP 40215A with DFMO or the bis(benzyl)polyamine analog MDL 27695 enhanced the activity of these drugs, leading to greater inhibition of promastigote growth [147]. As mentioned in Section 4.5, the bis(benzyl)polyamine analogs are potent inhibitors of both ODC and ADOMETDC [140], suggesting that the design of therapeutics that are dual inhibitors of targets within the polyamine pathway is a viable strategy.

It should be noted that substantial effort has been invested into the identification of inhibitors of the T. brucei ADOMETDC [173,174], and lead compounds could be useful for evaluation in Leishmania as well.

## 6. Spermidine Synthase (SPDSYN)

### 6.1. Enzyme Structure and Function

The *SPDSYN* gene was first identified in *L. donovani* in 2001 [175], and *SPDSYN* sequences from several *Leishmania* species are now available [114,115]. The *L. donovani*, *L. major*, and *L. braziliensis* SPDSYN proteins contain 300 amino acids, and sequence identity among the three *Leishmania* species ranges from 89 to 97% (Table 8). The *L. donovani* amino acid sequence is 45.6% and 45.4% identical to the mouse and human SPDSYN sequences, respectively (Table 8). Because SPDSYN is also essential in the human host, it is important to identify structural differences in order to minimize potential side effects. Although the crystal structure of the leishmanial SPDSYN is not available, the structures of the *T. cruzi* and *Plasmodium falciparum* SPDSYNs have been solved [77,176] and were used to model the leishmanial SPDSYN structure [150]. Moreover, it is feasible that the inhibition of SPDSYN in mammalian cells is not as detrimental as that of parasite SPDSYN because mammalian cells are able to convert spermine to spermidine and do not rely on trypanothione.

### 6.2. SPDSYN Gene Deletion Studies

Much like ODC, SPDSYN is an essential enzyme of the polyamine biosynthetic pathway and contributes to the infectious nature of *Leishmania* [108,175]. *L. donovani SPDSYN* gene deletion promastigotes are polyamine auxotrophs and not able to survive unless their growth media is supplemented with spermidine or the mutants are transfected with an *SPDSYN*-containing plasmid [108,175]. Culturing *SPDSYN* gene deletion mutants without spermidine supplementation leads to reduced spermidine and trypanothione levels but increased intracellular levels of putrescine and glutathione [127,175]. It is likely that a reduction in both spermidine and trypanothione levels causes cell death in these mutants. Furthermore, the *L. donovani SPDSYN* gene deletion mutants showed significantly reduced infectivity compared to wild-type parasites in murine infectivity studies, validating SPDSYN as a potential therapeutic target [108]. However, as discussed in the ODC section of this review, the phenotype of the *L. donovani SPDSYN* gene deletion mutants is less dramatic in vitro and in vivo when compared to the *ODC* gene deletion mutants. Although *SPDSYN* gene deletion promastigotes ultimately perish, the presence of putrescine allows them to enter a quiescent-like state and to persist for several weeks [127]. Persistence in *Leishmania* parasites has been observed in clinical studies and stimulated recent interest in understanding how parasites manage to subsist [21,25,27]. The *SPDSYN* gene deletion mutants may provide a model system to study cellular events required for parasites to enter a quiescent-like state. Although recent studies have highlighted the relevance of putrescine, it should be noted that spermidine levels are maintained at low but stable levels in both *SPDSYN* and *ODC* knockouts, demonstrating the importance of spermidine.

### 6.3. Inhibitors of SPDSYN

Screening of the natural product database ZINC identified the anthraquinone hypericin as an inhibitor of the leishmanial SPDSYN [177]. Hypericin is a natural compound derived from *Hypericum perforatum* (St. John’s wort), which is most commonly used as a dietary supplement for depression and topically to treat neurogenic pain. Hypericin exhibited a favorable docking score of −8.23 kcal/mol for the predicted structure of the parasite enzyme, while the docking score for the human enzyme was only −28 kcal/mol, suggesting a preference for the leishmanial SPDSYN [177]. Furthermore, the compound inhibited recombinant *L. donovani* SPDSYN and reduced spermidine and trypanothione levels in promastigotes, validating that the compound targets the parasite SPDSYN [177]. Hypericin inhibited promastigotes growth with an IC_50_ value of 18 µM and caused necrotic cell death [177]. Importantly, hypericin showed no toxicity against J774 macrophages [177], demonstrating selective toxicity toward parasites. Further studies were performed to investigate how the inhibition of SPDSYN causes cell death [178,179]. Supplementation with spermidine, but not trypanothione, rescued hypericin-treated parasites, demonstrating that spermidine has essential cellular functions beyond being a precursor for trypanothione formation and supporting redox metabolism [177]. Gene expression and proteomic analysis were performed with hypericin-treated parasites [178,179]. While changes in mRNA were most pronounced in genes coding for proteins involved in redox metabolism, the hypusine modification of eIF5A, DNA repair pathways, and autophagy, changes in the levels of proteins involved in protein synthesis, stress response, protein folding, and metabolic processes were predominantly observed [178,179].

Hypericin is also a naturally occurring photosensitizer due to its large chromophore system. Photodynamic therapy (PDT) is based on the principle that spectrum-specific light activates a compound and, together with oxygen, produces cytotoxic products. Hypericin is being tested as a PDT compound against cancer [180,181] as well as for the treatment of cutaneous leishmaniasis [182,183,184]. In vitro experiments with hypericin demonstrated that it was effective against intracellular *L. panamensis* parasites with and without light with EC_50_ values of 2.5 µM and 1.2 µM, respectively [183]. Infectivity studies in hamsters with a 0.5% hypericin topical cream with or without light exposure showed promising results with a cure of 3/8 and 6/8 hamsters, respectively [183]. No toxicity toward the animals was observed, and in vitro assays demonstrated that hypericin has wound-healing activities [183], which is an important aspect for the treatment of cutaneous leishmaniasis. These results demonstrate that hypericin is effective in vivo, presumably via its inhibition of parasite SPDSYN, and that treatment with light can potentiate its toxicity toward parasites. It is likely that the reduction in trypanothione levels via SPDSYN inhibition combined with increased oxidative stress due to PDT causes this increased antileishmanial activity, while the wound-healing capacity of hypericin contributes to positive outcomes [183]. Furthermore, the topical application of hypericin as a cream promises to be a better alternative compared to injections of traditional antileishmanial agents.

Aside from hypericin, other natural compounds have been identified that inhibit the leishmanial SPDSYN. Geraniol and Linalool are structural analogs of putrescine, and molecular docking studies showed high binding affinity and SPDSYN-inhibitor complex stability at the putrescine binding site [185]. DHC and BFPT are two natural compounds with dual inhibitory activity against ODC and SPDSYN but do not bind to the human SPDSYN [186]. A virtual screen demonstrated that the active sites of the human and leishmanial SPDSYN are sufficiently distinct to identify two compounds (pyridine and piperidin-1-ium analogs) that are predicted to bind to the leishmanial SPDSYN but not the human counterpart [150].

## 7. Polyamine Analogs

Two recent articles have reviewed classes of diamine and polyamine analogs and their efficacy against kinetoplastid parasites, including *Leishmania* spp. [53,187], and we will thus only briefly summarize these studies. A plethora of compounds have been synthesized that can be categorized broadly into diamine (putrescine analogs), triamine (spermidine analogs), tetramine (spermine analogs), longer polyamines, and cyclic derivatives. The modifications encompass a large variety of functional groups, including alkyl, acyl, aryl, and bisaryl groups as well as efforts to increase the lipophilicity of polyamine analogs to improve membrane permeability [53,187]. The mechanism of action is unknown for most of these molecules, although some compounds specifically inhibit polyamine biosynthetic enzymes (as discussed in the sections above) or trypanothione reductase, while others target superoxide dismutase, squalene synthase, arginine transporter, aminopurine transporter, or may cause mitochondrial damage [53,143,187,188,189,190]. Polyamine analogs may replace natural polyamines in the cell and disrupt numerous physiological processes, which is not surprising, considering that polyamines naturally have both specific and non-specific binding affinities to nucleic acids, proteins, lipids, and other cellular components, particularly those that are negatively charged. Unfortunately, compounds that were tested in vivo against kinetoplastids had only weak activities [187].

## 8. Polyamine Transport

In addition to a complete pathway for the biosynthesis of polyamines, *Leishmania* parasites are also capable of transporting polyamines and their precursors into the cell. *Leishmania* possess robust uptake activities for putrescine [191,192,193,194,195] and spermidine [192,194,195], as well as *S*-adenosylmethionine [196,197,198], L-arginine [199,200], and ornithine [72]. *Leishmania* parasites are also able to transport spermine, but they are unable to utilize spermine to fulfill their polyamine requirements [57]. Although inhibition of polyamine uptake by itself is likely to be of limited therapeutic value since *Leishmania* are able to synthesize polyamines, dual targeting approaches, where an inhibitor against the polyamine biosynthetic pathway is combined with a transport inhibitor, have shown promise against tumor cells, as well as in rodent tumor models [201,202,203,204]. Moreover, transporters for polyamines and polyamine precursors will likely be of significance in the delivery of drugs directed against the polyamine pathway.

### 8.1. Arginine Transport

*Leishmania* are auxotrophic for arginine and possess a high affinity transporter AAP3 that, like other amino acid and polyamine transporter sequences characterized in *Leishmania*, belongs to the amino acid auxin permease family TC 2.A.18 [200,205]. AAP3 is present in the chromosome as two tandemly arranged gene sequences that are virtually identical at the amino acid level [200]. However, in *L. donovani*, only one of these gene copies, *LdAAP3.2*, is regulated in response to changes in host arginine levels and notably, when deleted from the *L. donovani* genome, it impeded the infection of host macrophages and BALB/C mice [91,200]. This suggests that targeting of this transporter may be of therapeutic value, particularly considering that higher levels of host arginine have been linked to increased nitric oxide production and reduced parasite survival [80,102,206].

### 8.2. Ornithine Transport

The genes encoding ornithine transporters in *Leishmania* have yet to be uncovered; however, in *T. brucei* two coding sequences, TbAAT10 and TbAAT2-4, have been identified as L-ornithine and L-ornithine/L-histidine transporters, respectively [207], and in *T. cruzi* ornithine uptake is mediated by a high affinity L-arginine/low affinity L-ornithine transporter, TcCAT1.1 [208]. The closest paralogs to these sequences in *Leishmania* spp. are AAP3 (arginine transporter) and aATP11. However, L-ornithine does not appear to compete for arginine transport by AAP3 [200], and the substrate specificity of aATP11 has yet to be determined. Of note, aATP11 mRNA is upregulated in *L. amazonensis ARG* gene deletion mutants [209], where the conversion of arginine to ornithine is disrupted.

DFMO is a structural analog of L-ornithine; however, it is unclear whether DFMO gains access by the same transport system as L-ornithine in *Leishmania*. In *T. brucei*, DFMO is transported into the parasite by a separate neutral amino acid transporter, TbAAT6 [210,211] (the homolog for which is AAP24 in *Leishmania*, a proline-alanine transporter). Moreover, DFMO does not appear to compete against ornithine, arginine, or putrescine for uptake into *T. brucei* [212].

### 8.3. Putrescine Transport

The POT1 putrescine transporter (AAT21) was initially identified within the genome of *L. major* [194], but it has subsequently been identified within the sequenced genomes of several other *Leishmania* species [114,115] as well as in other related trypanosomatids [213,214]. Heterologous expression of the *L. major* POT1 in oocytes and *T. brucei* suggest that it is a high-affinity putrescine transporter (K_m_ 7–10 µM) that also has specificity toward spermidine (K_m_ ≈ 10 µM) [194]. Putrescine transport appears to be regulated by changes in the extracellular and intracellular putrescine environment. High levels of exogenous putrescine repressed transport, while the inhibition of ODC by DFMO caused a decrease in the intracellular putrescine concentration and enhanced the uptake of putrescine from the culture media [193,195]. Although the physiological significance of polyamine uptake via POT1 has yet to be established in terms of its contribution to *Leishmania* virulence, POT1 is the target of various therapeutics. Pentamidine, an antileishmanial diamidine drug, which shares some structural similarity to putrescine and spermidine, is a potent inhibitor of putrescine uptake via POT1 [192,194]. DAB, a putrescine analog and ODC inhibitor, also inhibits putrescine uptake in *Leishmania* parasites, although prolonged exposure to DAB appears to lead to enhanced putrescine uptake [143]. Likewise, GAPA, an agmatine analog and ODC inhibitor, inhibits putrescine uptake into the parasite [142].

### 8.4. Spermidine Transport

While at least a portion of spermidine uptake into *Leishmania* is mediated by the POT1 transporter, it is likely that these parasites possess an additional high-affinity spermidine transporter, since spermidine uptake was only partially competed for by exogenous putrescine [192,194]. Notably, the uptake of spermidine into the parasite, but not putrescine, was also partially competed for by spermine [193,195], suggesting that POT1-independent spermidine transport into *Leishmania* is via a transporter with a shared specificity for spermine.

### 8.5. S-adenosylmethionine Transport

Unlike mammalian cells that rely on synthesis for the provision of *S*-adenosylmethionine, *Leishmania* are able to salvage *S*-adenosylmethionine from the host via a high-affinity transport system as well as synthesize this metabolite from methionine and ATP [196,197,198,215,216]. The *Leishmania S*-adenosylmethionine transporter (AdoMetT1) is a member of the folate-biopterin transporter family [197]. The substrate-specificity profile for AdoMetT1 indicates that it is highly specific for *S*-adenosylmethionine and has no affinity for the structurally related metabolites, methionine, ornithine, or adenine and adenosine [197,198,216]. However, *S*-adenosylmethionine transport via AdoMetT1 is potently competed by the drug sinefungin, which is a structural analog of *S*-adenosylmethionine [196,197,216]. Deletion of the *AdoMetT1* gene from the genome of *L. infantum* confirms that AdoMetT1 is the sole route of uptake for *S*-adenosylmethionine and sinefungin [197], but the effect of this deletion on parasite growth and infectivity in macrophages is not known.

## 9. Drug Resistance and Polyamines

In recent years, drug resistance and therapeutic failure have become a significant issue in the treatment of leishmaniasis, particularly for mainstay antimonial drugs, curtailing their utility in many regions where leishmaniasis is endemic [217,218,219,220]. While drug resistance is a complex phenomenon, changes in parasite antioxidant pathways, particularly in trypanothione metabolism, have commonly been linked to drug resistance in *Leishmania* [221,222,223,224,225,226,227,228,229,230,231]. Trypanothione, the major intracellular thiol in these parasites, is the cornerstone of parasite oxidant defense, and therefore, changes within parasite polyamine levels are of particular interest due to their connection to trypanothione biosynthesis.

Alterations in trypanothione metabolism have been well-documented in both clinical and laboratory-derived isolates resistant to antimonial drugs [227,232,233], and additionally, changes in gene expression within the polyamine pathway have frequently been observed [221,223,226,229,234,235,236]. Increased levels of *ODC* mRNA expression have been observed in clinical isolates of *L. donovani*, *L. infantum*, and *L. braziliensis*, which exhibit resistance to pentavalent antimonial drugs [221,223,234], and the amplification of the *ODC* gene locus has been detected in clinical isolates of *L. donovani* resistant to sodium antimony gluconate [226]. However, the link between increased ODC expression and antimony resistance is not straightforward. Comparison of *ODC* mRNA levels by quantitative real-time PCR in several *L. tropica* isolates that were either sensitive or resistant to pentavalent antimonial drugs revealed no significant change [233]. Additionally, an analysis of *L. donovani* intracellular amastigotes cultured from clinical isolates exhibiting either sensitivity or resistance to pentavalent antimonial drugs indicated that *ODC* mRNA levels were 2–3-fold lower in the resistant isolates. However, it should be noted that mRNA levels in general were lower in the amastigote stage of the resistant isolates in these analyses [234]. In contrast, overexpression of the *ODC* gene from a plasmid in an experimental model of *L. (Vianna) guyanensis* rendered the cells 2 to 4-fold resistant to trivalent antimony, and this could be partially reversed by the pharmacological inhibition of ODC by the drug DFMO [235]. Changes within other components of the polyamine pathway in antimony-resistant isolates have not been widely reported in the literature, although in one study where changes in the proteome of antimony-resistant clinical isolates of *L. panamensis* were assessed, protein levels for *S*-adenosylmethionine synthetase appeared to be upregulated in the resistant parasites [229].

Alterations in polyamine metabolism have been reported in the clinical resistance to other standard antileishmanial drugs. Metabolomic analyses undertaken on laboratory-derived strains of *L. donovani*, which exhibited either a sensitive or resistant phenotype to miltefosine, indicated a perturbation of arginine, ornithine, and *S*-adenosylmethionine levels in the sensitive strain, and increased levels of arginine, ornithine, *S*-adenosylmethionine, and spermidine in the resistant strain upon exposure to miltefosine [237]. Laboratory-derived, pentamidine-resistant strains of *L. donovani* and *L. amazonensis* also showed an increase in intracellular arginine and ornithine pools, although putrescine levels were decreased by 6-fold in the resistant strain. This was attributable to a decrease in ODC protein and activity as well as an impairment of putrescine uptake from the culture media [238,239].

The selection of laboratory-derived mutants resistant to DFMO and other drugs targeting the polyamine pathway has also been described [134,135,137,142,155,228,240]. Indeed, it appears that resistance to DFMO may be relatively easy to select for in culture [134,137,228,240], due to amplification events at the *ODC* gene locus [113,134,136], which could preclude the therapeutic targeting of ODC. It should be mentioned, however, that it is unclear what effect amplification of the *ODC* gene may exert on parasite fitness and survival in clinical isolates, although there is at least one report of *ODC* gene amplification in a clinical isolate displaying antimony resistance [226]. Moreover, *ODC* overexpressing strains, despite displaying resistance to DFMO and other drugs that target this activity, retain sensitivity to some inhibitors of the downstream polyamine pathway enzymes [135,137]. Ultimately, a combination of chemotherapeutic approaches, where a polyamine pathway inhibitor is paired and potentiated with another antileishmanial drug, or even a polyamine transport inhibitor [201,202,203,204], may hold promise as a therapeutic approach. Similarly, the inhibition of polyamine and trypanothione synthesis might restore the sensitivity to antimonial drugs in some antimony-resistant isolates [235] or even prevent the development of antimony resistance. Such combination chemotherapeutic approaches have yet to be fully evaluated in these parasites.

## 10. Discussion

There are several lines of evidence that promote the polyamine pathway in *Leishmania* as a promising therapeutic target. Polyamines are essential for proliferation, survival, and infectivity, and significant differences exist in the metabolic pathways of host and parasite. Moreover, *L. donovani ODC* and *SPDSYN* gene deletion mutants exhibit profoundly reduced infectivity phenotypes in mice [107,108], and ODC, SPDSYN, and ADOMETDC inhibitors demonstrate efficacy in vitro and in vivo, boosting the approach of therapeutically targeting this pathway.

A significant reduction in parasite loads in rodents has been observed during treatment with the ODC inhibitors DFMO (Table 3) and MDL 27695 [140,145,146] and the natural compound hypericin, which is an inhibitor of SPDSYN [183]. The results with DFMO, however, indicate that differences in susceptibility may exist within the various *Leishmania* species (Table 2 and Table 3). Differences in transport of DFMO and/or salvage of putrescine could contribute to the variable efficacy, and future studies are necessary to shed light on the species-dependent disparities in susceptibility.

While the crystal structure of the *Leishmania* polyamine biosynthesis enzymes, with the exception of the *L. mexicana* ARG [241,242], have not been elucidated, the structures for the *T. brucei* ODC, ADOMETDC, and ADOMETDC/prozyme heterodimer, and the *T. cruzi* and *Plasmodium falciparum* SPDSYN [32] are available and have been used for modeling of the *Leishmania* enzyme structures and in silico drug screening and discovery of inhibitors [116,117,118,149,150,151,177]. All three polyamine pathway enzymes, ODC, ADOMETDC, SPDSYN, are arguably druggable targets because small molecules that inhibit the enzymes have been identified.

The inhibition of any of the polyamine biosynthetic enzymes could potentially be circumvented by drug-resistance mechanisms, including a gene amplification event. Monotherapy with current antileishmanial drugs has demonstrated that the emergence of drug resistance is inevitable [13,15,217,218,219,220]. It is thus not surprising that combination therapy against leishmaniasis has recently received more attention [13,19,20]. Combination therapies of established antileishmanial agents with polyamine biosynthetic enzymes represent a new avenue for exploration.

Substantial differences in the polyamine biosynthetic pathways of the mammalian host and parasite have been revealed. *Leishmania* lack a back-conversion pathway from spermine to spermidine and putrescine, and they do not synthesize or utilize spermine (a dominant polyamine in the mammalian host). Furthermore, ODC and ADOMETDC are stable enzymes in the parasite, while the host counterparts are rapidly turned over, and thus, any covalently bound inhibitor would be more impactful on protein function in parasites. Structural differences also exist between the *Leishmania* polyamine biosynthetic enzymes and those of the mammalian host. For example, the leishmanial ODCs possess a unique N-terminal extension that is essential for their function and may allow for the specific targeting of these enzymes over the human ODC. Furthermore, the trypanosomatid ADOMETDC is uniquely activated by the formation of a heterodimer with prozyme, which is a catalytically dead paralog of the parasite ADOMETDC. These differences in the polyamine biosynthetic pathway steps, enzyme structures, and half-lifes should facilitate the development of therapeutic strategies that specifically target the parasite.

Polyamines are essential for the proliferation and survival of *Leishmania* promastigotes and intracellular amastigotes; however, the specific molecular mechanisms by which they facilitate these processes are still largely unknown. A comparison of the starvation phenotype of *L. donovani* gene deletion mutants demonstrates that putrescine is not merely a precursor for spermidine formation, as previously postulated, but it is especially important for proliferation in *Leishmania* promastigotes [69,127]. MDL 27695 and related inhibitors of ODC or ADOMETDC showed a decrease in protein synthesis [148], and inhibition of SPDSYN with hypericin resulted in translational arrest [177]. Spermidine is essential for the hypusination and activation of eIF5A, and it is likely that the reduced processing of eIF5A also contributes to the inhibition of protein synthesis. The effect of hypericin treatment on global gene expression suggests that spermidine depletion furthermore affects DNA repair, autophagy, and oxidative stress management. Polyamines are crucial as precursors for the synthesis of trypanothione, and the inhibition or deletion of ODC, SPDSYN, or ADOMETDC leads to reduced levels of trypanothione, which is a thiol that is essential for combatting oxidative stress [57,155,175,177]. Not surprisingly, an increase in reactive oxygen species has been observed in parasites when polyamine pathway enzymes were inhibited [116,177]. Together, these studies provide some insight into the functions of polyamines as key metabolites for proliferation, protein synthesis, and oxidative stress management, but more studies are needed to elucidate their specific roles.

Host–parasite interactions and the unique intracellular niche in which *Leishmania* parasites exist need to be taken into consideration for the development of any therapeutic strategy. The formulation of polyamine enzyme inhibitors, or for that matter any anti-parasitic compound, to better target the phagolysosome and tissue lesions (that are often comprised of granulomas, which are refractory to drug delivery) promises to increase efficacy. It is well established that *Leishmania* parasites modulate host macrophage metabolism and the immune response. While successful *Leishmania* infections are known to increase the amount and activity of host arginase, the impact on polyamine synthesis has not been directly investigated. Indeed, despite the common (mis)conception that an increase in arginase activity will increase salvageable host polyamine levels, putrescine and spermidine levels are typically present only at low levels in differentiated cells, while spermine (which *Leishmania* cannot utilize) is dominant [57,99,109,110]. The significantly reduced infectivity phenotype of the *L. donovani ODC* and *SPDSYN* gene deletion mutants [107,108] also suggests that putrescine and spermidine are not available for parasite salvage under physiological conditions. Yet, dietary uptake or variable local tissue concentrations of polyamines may need to be taken into consideration. A dual strategy of inhibiting polyamine biosynthesis and transport has shown some promise as a model for the treatment of certain cancers [201,202,203,204], and it might be duplicated for resolving *Leishmania* infections. Further investigation of the structure–activity relationship for polyamine transport in *Leishmania* parasites, as well as a better understanding of host–parasite polyamine interaction, will help resolve these outstanding questions.

## 11. Conclusions

The polyamine biosynthetic pathway is a promising therapeutic target in *Leishmania* parasites. Hence, further development of compounds, potential combination therapies, and investigations into the functions of polyamines as well as host–parasite interactions are well warranted and open exciting new areas for exploration.

## Figures and Tables

**Figure 1 medsci-10-00024-f001:**
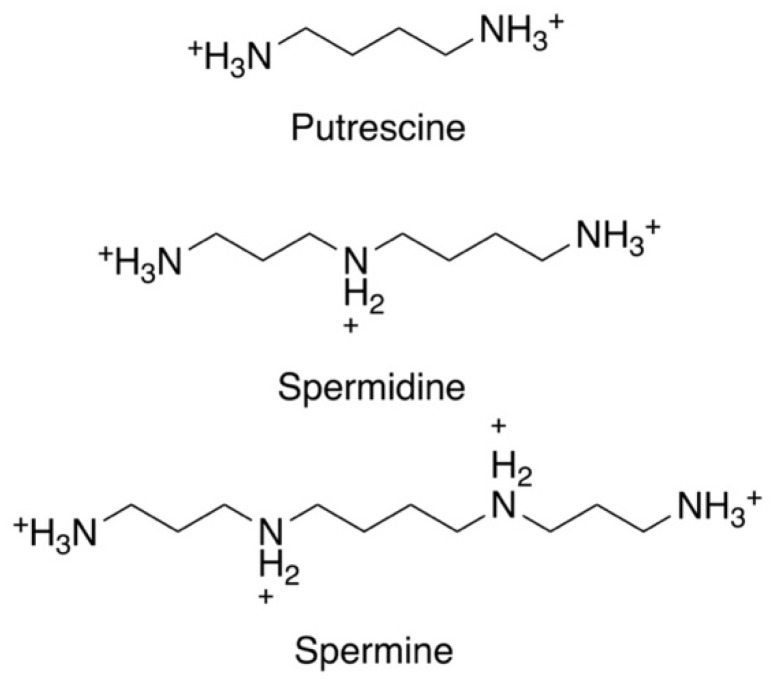
Chemical structures of the three main polyamines: putrescine, spermidine, and spermine.

**Figure 2 medsci-10-00024-f002:**
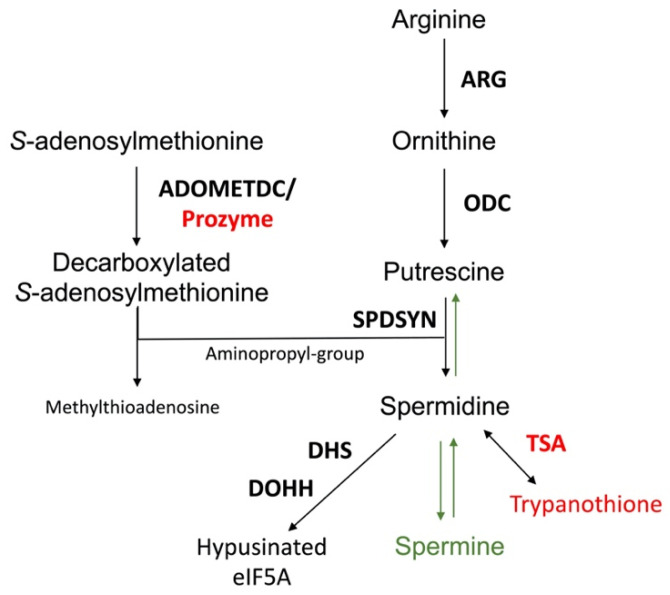
The polyamine biosynthetic pathway. Enzymes are shown in bold. The polyamine biosynthetic enzymes are arginase (ARG), ornithine decarboxylase (ODC), spermidine synthase (SPDSYN), and *S*-adenosylmethionine decarboxylase (ADOMETDC), the latter depicted with prozyme. Downstream reactions are catalyzed by trypanothione synthetase/amidase (TSA), a bifunctional enzyme that forms trypanothione, and by deoxyhypusine synthase (DHS) and deoxyhypusine hydroxylase (DOHH) that in consecutive reactions catalyze the hypusination and activation of eIF5A. The enzymes and metabolites unique to trypanosomatids, prozyme, TSA, and trypanothione, are shown in red, while the synthesis of spermine and the back-conversion of spermidine to putrescine, present only in mammalian cells and not in *Leishmania*, are denoted in green.

**Figure 3 medsci-10-00024-f003:**
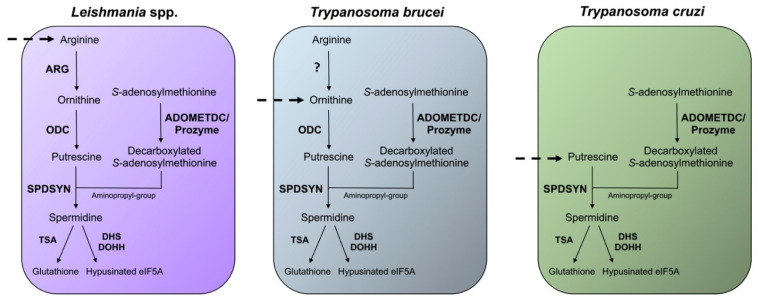
Differences in the polyamine biosynthetic pathways between trypanosomatids. The polyamine pathways of *Leishmania* spp., *T. brucei*, and *T. cruzi* are depicted. Enzymes are shown in bold. The polyamine biosynthetic enzymes spermidine synthase (SPDSYN), *S*-adenosylmethionine decarboxylase (ADOMETDC) with prozyme, trypanothione synthase (TSA), deoxyhypusine synthase (DHS), and deoxyhypusine hydroxylase (DOHH) are present in all three trypanosomatids. Ornithine decarboxylase (ODC) is missing in *T. cruzi*, and only *Leishmania* spp. appear to contain an active arginase (ARG). Dashed arrows represent the indispensable transport of polyamines or polyamine precursors.

**Figure 4 medsci-10-00024-f004:**
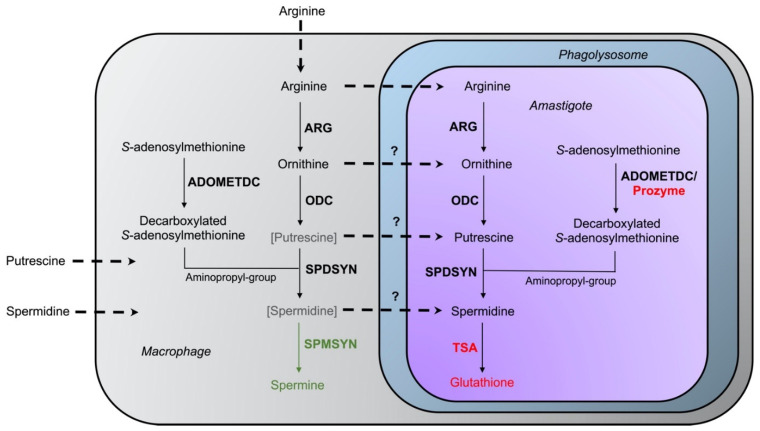
Interaction of host–parasite polyamine pathways. The enzymes arginase (ARG), ornithine decarboxylase (ODC), spermidine synthase (SPDSYN), and *S*-adenosylmethionine decarboxylase (ADOMETDC) are present in both host and parasite, while the enzyme spermine synthase (SPMSYN), displayed in green, can only be found in the host. The enzyme trypanothione synthetase/amidase (TSA) and prozyme, shown in red, are unique to the parasite. Putrescine and spermidine are bracketed and depicted in gray in the macrophage to indicate that only low amounts may be present in host cells. The uptake of arginine, ornithine, putrescine, and spermidine is represented in the dotted arrows. The question marks denote that the extent of transport of ornithine, putrescine, and spermidine into the phagolysosome and intracellular amastigote under physiological conditions is unclear.

**Figure 5 medsci-10-00024-f005:**
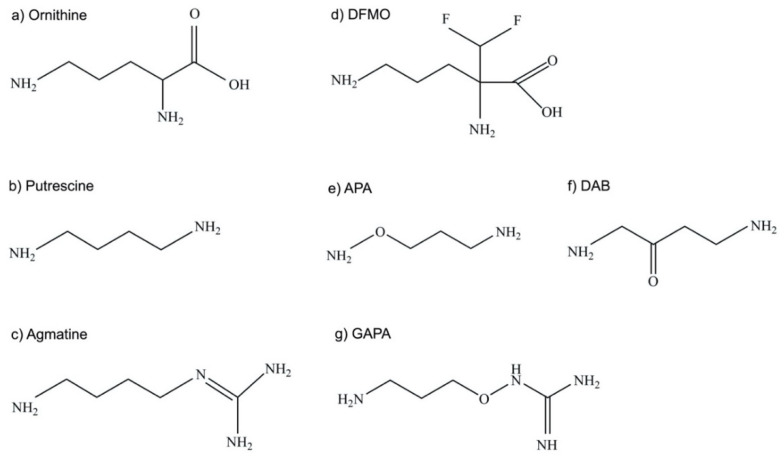
Structures of polyamine pathway metabolites and select ODC inhibitors. Structures of the metabolites (**a**) ornithine, (**b**) putrescine, and (**c**) agmatine are shown, as well as the following ODC inhibitors: (**d**) D, L-α-difluoromethylornithine (DFMO), the putrescine derivatives (**e**) 3-aminooxy-1-aminopropane (APA) and (**f**) 1,4-diamino-2-butanone (DAB), and the agmatine derivative (**g**) gamma-guanidinooxy propylamine (GAPA).

**Table 1 medsci-10-00024-t001:** Amino acid sequence identity in percent among ODC proteins from various species.

	*L. donovani*	*L. major*	*L. braziliensis*	*T. brucei*	*Mus musculus*	*Homo sapiens*
** *L. donovani* **	100%	91.5%	64.2%	24.5%	25.1%	23.9%
** *L. major* **		100%	63.6%	24.1%	24.7%	23.6%
** *L. braziliensis* **			100%	26.3%	25.3%	25.9%
** *T. brucei* **				100%	58.6%	59.7%
** *Mus musculus* **					100%	90.7%
** *Homo sapiens* **						100%

Sequence alignments were performed with Clustal Omega, Needle (EMBOSS), which aligns two sequences along their entire length. Sequences: *L. donovani* BPK282A1 LdBPK_120105.1, *L. major* Friedlin LmjF.12.0280, *L. braziliensis* MHOM/BR/75/M2904 LbrM.12.0300, *T. brucei* TREU927 Tb927.11.13730, *Mus musculus* NP_038642.2, *Homo sapiens* P11926. A multi-sequence alignment of the ODC sequences is shown in Appendix A.

**Table 2 medsci-10-00024-t002:** In vitro efficacy of DFMO.

*Leishmania* Species	Inhibition of Parasites Growth In Vitro	Reference
** *L. donovani* **	EC_50_ 40 µM in axenic amastigotes; DFMO protected macrophages from infection	[107]
	No growth inhibition of promastigotes, but EC_50_ 50 µM in intracellular amastigotes	[59]
	5 mM effectively inhibited promastigote growth	[138]
	EC_50_ 30 µM in promastigotes	[58]
	EC_50_ 125 µM in promastigotes	[137]
** *L. infantum* **	EC_50_ 38 µM in promastigotes	[133]
** *L. mexicana* **	No effect in promastigotes	[138]
	10 mM DFMO suppressed promastigote growth after seven passages or 28 days of growth	[76]
	5 mM DFMO suppressed promastigote growth after three passages or 10 days of growth	[134]
** *L. major* **	No effect in promastigotes	[138]
** *L. braziliensis* ** ** *guyanensis* **	5 mM effectively inhibited promastigote growth	[138]

EC_50_: half maximum effective concentration.

**Table 3 medsci-10-00024-t003:** In vivo efficacy of DFMO.

*Leishmania* Species	Type of Rodent	Amount, Administration, and Efficacy	References
** *L. donovani* **	BALB/c mice	2% in drinking water for 3 weeks reduced liver parasite burden by 93% but had no effect on parasite numbers in the spleen	[111]
	BALB/c mice	1% and 3% in drinking water for 7 days suppressed liver burden by 16% and 53%, respectively	[138]
	Golden Hamster	2% in drinking water 2 days after infection and continued for 4 days reduced infection in liver and spleen by 90% and 99%, respectively	[140]
** *L. infantum* **	BALB/c mice	100 mg/kg subcutaneous for 5 days and 200 mg/kg subcutaneous for 42 days reduced infection in liver by 85% and 98%, respectively	[139]
** *L. mexicana* **	BALB/c mice	2% and 4% in drinking water for 3 weeks reduced infections by 12% and 20%	[138]
** *L. braziliensis* ** ** *guyanensis* **	BALB/c mice	2, 4, and 5% in drinking water reduced lesion size by 100%, 43%, and 81%, respectively	[138]

**Table 4 medsci-10-00024-t004:** Efficacy of APA and GAPA in *L. donovani*.

Compound	Inhibition of Recombinant *L. donovani* ODC (K_i_)	Inhibition of Promastigotes (IC_50_)	Inhibition of Intracellular Amastigotes (IC_50_) in J774A.1Macrophages	References
**APA**	1.0 nM	42 ± 8 µM	5 ± 2.0 µM	[59,118,141]
**GAPA**	60 µM	36 ± 7.0 µM	9 ± 1.0 µM	[141,142]

GAPA: gamma-guanidinooxy propylamine or 1-guanidinooxy-3-aminopropane. APA: 3-aminooxy-1-aminopropane. K*i*: inhibition constant. EC_50_: half maximum effective concentration.

**Table 5 medsci-10-00024-t005:** Amino acid sequence identity in percent among ADOMETDC proteins from various species.

	*L. donovani*	*L. major*	*L. braziliensis*	*T. brucei*	*T. cruzi*	*Mus musculus*	*Homo sapiens*
** *L. donovani* **	100%	97.6%	88.5%	61.8%	70.9%	26.2%	26.5%
** *L. major* **		100%	88.5%	61.9%	70.4%	26.2%	26.5%
** *L. braziliensis* **			100%	61.9%	70.1%	27.0%	27.0%
** *T. brucei* **				100%	68.3%	26.8%	26.9%
** *T. cruzi* **					100%	25.9%	25.9%
** *Mus musculus* **						100%	98.2%
** *Homo sapiens* **							100%

Sequence alignments were performed with Clustal Omega, Needle (EMBOSS), which aligns two sequences along their entire length. Sequences: *L. donovani* BPK282A1 LdBPK_303150.1, *L. major* Friedlin LmjF.30.3110, *L.*
*braziliensis* MHOM/BR/75/M2903 LBRM2903_300037800, *T. brucei* TREU927 Tb927.6.4410, *T. cruzi* CL TcCL_ESM01038, *Mus musculus* NP_033795.1, and *Homo sapiens* NP_001625.2. A multi-sequence alignment of the ADOMETDC sequences is shown in Appendix A.

**Table 6 medsci-10-00024-t006:** Amino acid sequence identity in percent among prozyme proteins from various trypanosomatid species.

	*L. donovani*	*L. major*	*T. brucei*	*T. cruzi*
** *L. donovani* **	100%	89%	40.8%	40.2%
** *L. major* **		100%	40.4%	41.9%
** *T. brucei* **			100%	48.5%
** *T. cruzi* **				100%

Sequence alignments were performed with Clustal Omega, Needle (EMBOSS), which aligns two sequences along their entire length. Sequences: *L. donovani* BPK282A1 LdBPK_303160.1, *L. major* Friedlin LmjF.30.3120, *T. brucei* Lister strain 427 Tb427.06.4470, *T. cruzi* CL Brener Esmeraldo-like TcCLB.509167.110. A multi-sequence alignment of the prozyme sequences is shown in Appendix A.

**Table 7 medsci-10-00024-t007:** Comparison of amino acid sequence identities of ADOEMTDC and prozyme within each species.

*L. donovani* ADOMETDC versus *L. donovani* Prozyme	*L. major* ADOMETDC versus *L. major* Prozyme	*T. brucei* ADOMETDC versus *T. brucei* Prozyme	*T. cruzi* ADOMETDC versus *T. cruzi* Prozyme
**24.4%**	23.1%	23.2%	26.3%

Sequence alignments were performed with Clustal Omega, Needle (EMBOSS), which aligns two sequences along their entire length. Sequences: ADOMETDC: *L. donovani* BPK282A1 LdBPK_303150.1, *L. major* Friedlin LmjF.30.3110, *L.*
*braziliensis* MHOM/BR/75/M2903 LBRM2903_300037800, *T. brucei* TREU927 Tb927.6.4410, *T. cruzi* CL TcCL_ESM01038, *Mus musculus* NP_033795.1, *Homo sapiens* NP_001625.2; Prozyme: *L. donovani* BPK282A1 LdBPK_303160.1, *L. major* Friedlin LmjF.30.3120, *T. brucei* Lister strain 427 Tb427.06.4470, and *T. cruzi* CL Brener Esmeraldo-like TcCLB.509167.110.

**Table 8 medsci-10-00024-t008:** Amino acid sequence identity in percent among SPDSYN proteins from various species.

	*L. donovani*	*L. major*	*L. braziliensis*	*T. cruzi*	*T. brucei*	*Mus musculus*	*Homo sapiens*
** *L. donovani* **	100%	96.7%	89.0%	65.6%	67.0%	45.6%	45.4%
** *L. major* **		100%	88.7%	65.9%	66.0%	44.3%	44.1%
** *L. braziliensis* **			100%	65.9%	65.3%	44.3%	42.8%
** *T. cruzi* **				100%	71.8%	43.1%	42.3%
** *T. brucei* **					100%	44.6%	43.8%
** *Mus musculus* **						100%	94.7%
** *Homo sapiens* **							100%

Sequence alignments were performed with Clustal Omega, EMBOSS Needle. Sequences: *L. donovani* BPK282A1 LdBPK_040570, *L. major* Friedlin LmjF.04.0580, *L. braziliensis* MHOM/BR/75/M2904 LbrM.04.0630, *T. cruzi* Dm28c 2017 BCY84_20019, *Mus musculus* NP_033298.1, *Homo sapiens* NP_003123.2. A multi-sequence alignment of the SPDSYN sequences is shown in Appendix A.

## Data Availability

Publicly available data sets were analyzed in this study. Trypanosomatid sequences were retrieved from the TriTrypDB Kinetoplastid Informatics Resources (Release 55 2 December 2021) (https://tritrypdb.org/tritrypdb/app, accessed on 1 October 2021), and mouse and human sequences were retrieved from the NCBI Protein Database (https://www.ncbi.nlm.nih.gov/protein, accessed on 1 October 2021).

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
