# Peer review of "Polyamine Metabolism in Leishmania Parasites: A Promising Therapeutic Target"

_medsci, 2022, doi:10.3390/medsci10020024_

Round 1

Reviewer 1 Report

The review is well done and very interesting. The only thing I found is:

Line 213: replace AND with AN (before essential).

Best regards.

Author Response

Thank you for the positive review. We have corrected the word in line 213.

Reviewer 2 Report

This review paper is devoted to polyamines for the treatment of leishmaniasis and considers enzymes for the synthesis and metabolism of polyamines as therapeutic molecular targets. The authors did not focus only on Leishmania, but also provided data on other related species, including Trypanosoma spp. The review is quite extensive, and includes both a discussion of the results of structural crystal analysis, in vitro enzyme inhibitors, and studies on the inhibition of growth and reproduction of leishmania and trypanosomides in vivo. In my opinion, the review will be of interest to readership of the Journal and experts in the field of Drug Development against parasitoses.

Author Response

Thank you for the favorable feedback.

Reviewer 3 Report

The paper “Polyamines in Leishmania parasites: a promising therapeutic target” by Carter et al.  focuses on the main polyamine biosynthetic enzymes: ornithine decarboxylase (ODC), S-adenosylmethionine decarboxylase (ADOMETDC) and spermidine synthase (SPDSYN), and emphasizes recent discoveries that advance these enzymes as potential therapeutic targets against Leishmania parasites. The paper is well written and there are not  recent review to our knowledge, on the polyamine pathway, focused on the aforementioned enzymes. For this reason, I recommend the paper for publication in Medical Sciences. I have only minor requests

Minor points

  1. In the paragraph 4.3 the authors discuss the efficacy of the ornithine analog DFMO as ODC Inhibitor in Leishmania species. The authors report that there are discrepancies of DFMO efficacy among Leishmania species: L. donovani and L. infantum promastigotes are highly susceptible to DFMO, while L. mexicana and L major are resistant to this compound. This is an interesting point, but the authors report as reference, for the inhibition experiment on Leishmania major and L. brasiliensis on Table 2, a book chapter (ref. 138: Keithly, J. S., Fairlamb, A.H. "Inhibition of Leishmania Species by Α-Difluoromethylornithine." In Leishmaniasis, edited by 1312 D. T. Hart. Boston, MA.: NATO ASI Series, 1989. 1313). I suppose there are  original experimental papers reporting these data that should be cited.
  2. The authors cited inhibitors of the three enzymes. It would be important for the reader to have a table summarizing the data discussed in the paper. Please add a table, reporting the inhibitor names, formulas, Ki on the target enzyme, promastigote IC50 and the  corresponding references.

Author Response

Response to minor points:

  1. The cited book chapter is indeed the original reference that states the experimental data.
  2. We like this idea and have attempted to compile such a table. However, the different papers report efficacy of the various inhibitors in substantially different manners. For example, some papers measure Ki values of recombinant enzymes or cell extracts, others merely state general enzyme inhibition, IC50s are not available for all inhibitors. Moreover, a useful table should also include efficacy against intracellular amastigotes in vitro and in vivo. Such a table would become very complicated and difficult to read and interpret. Thus, we decided to not include the table and believe that the narrative in our review is sufficient.

Reviewer 4 Report

The authors present a very complete review about the polyamines pathway as target for antileishmanial drug design. In general, the manuscript has scientific soundness and will be of interest to those working in the search of new agents against trypasomatids and particularly against leishmania sp. Just a couple of suggestions before its acceptance for publicaction.

1.    The title should be modified because Polyamines are not the target, the target is the polyamines pathway or enzymes from polyamines pathway.
2.    The discussion section should be deleted because the information stated there is almost the same described in the other sections of the manuscript. Instead, a section of “future perspectives” would be desirable.

Author Response

  1. We have modified the title to "Polyamine metabolism in Leishmania parasites: a promising therapeutic target"
  2. We have carefully considered this suggestion to delete the discussion, but believe that this section is a valuable summary and cohesive discussion of the major points of the individual sections. Because the review is extensive, we find it important to have this final piece that brings together major points. We are also aware that some readers may not read an entire manuscript but rather focus on the introduction and discussion.